# Evaluation of two red cell inclusion staining methods for assessing spleen function among sickle cell disease patients in North-East Nigeria

**Adama I. Ladu**[1,2]*, **Ngamarju A. Satumari**[3], **Aisha M. Abba**[2], **Fatima A. Abulfathi**[2], **Caroline Jeffery**[4,5], **Adekunle Adekile**[6], **Imelda Bates**[1]

1 Department of International Public Health, Liverpool School of Tropical Medicine, Liverpool, United Kingdom, 2 Department of Haematology, Faculty of Basic Clinical Sciences, University of Maiduguri, Maiduguri, Borno State, Nigeria, 3 Department of Medical Laboratory Science, Faculty of Allied Health Sciences, University of Maiduguri, Maiduguri, Borno state, Nigeria, 4 Department of Clinical Infection, Microbiology and Immunology, Faculty of Health and Life Sciences, University of Liverpool, Liverpool, United Kingdom, 5 Department of Health Data Science, University of Liverpool, Liverpool, United Kingdom, 6 Department of Paediatrics, Faculty of Medicine, Kuwait University, Kuwait, Kuwait

* adamaisahladu@gmail.com, adama.ladu@lastmed.ac.uk

**Data Availability Statement:** The data underlying this article are available in the manuscript, figures,

## Abstract

The loss of splenic function is associated with an increased risk of infection in sickle cell disease (SCD); however, spleen function is rarely documented among SCD patients in Africa, due partly to the non-availability of sophisticated techniques such as scintigraphy. Methods of assessing splenic function which may be achievable in resource-poor settings include counting red blood cells (RBC) containing Howell Jolly Bodies (HJB) and RBC containing silver-staining (argyrophilic) inclusions (AI) using a light microscope. We evaluated the presence of HJB—and AI—containing RBC as markers of splenic dysfunction among SCD patients in Nigeria. We prospectively enrolled children and adults with SCD in steady state attending outpatient clinics at a tertiary hospital in North-East Nigeria. The percentages of HJB—and AI-containing red cells were estimated from peripheral blood smears and compared to normal controls. There were 182 SCD patients and 102 healthy controls. Both AI- and HJB-containing red cells could be easily identified in the participants blood smears. SCD patients had a significantly higher proportion of red cells containing HJB (1.5%; IQR 0.7%–3.1%) compared to controls (0.3%; IQR 0.1%–0.5%) ($P < 0.0001$). The AI red cell counts were also higher among the SCD patients (47.4%; IQR 34.5%–66.0%) than the control group (7.1%; IQR 5.1%–8.7%) ($P < 0.0001$). The intra-observer reliability for assessment of HJB- ($r = 0.92$; $r^2 = 0.86$) and AI- containing red cells ($r = 0.90$; $r^2 = 0.82$) was high. The estimated intra-observer agreement was better with the HJB count method (95% limits of agreement, -4.5% to 4.3%; $P = 0.579$). We have demonstrated the utility of light microscopy in the assessment of red cells containing—HJB and AI inclusions as indices of splenic dysfunction in Nigerian SCD patients. These methods can be easily applied in the routine evaluation and care of patients with SCD to identify those at high risk of infection and initiate appropriate preventive measures.

and tables. Further request can be addressed to the corresponding author.

**Funding:** The authors received no specific funding for this work.

**Competing interests:** The authors have declared that no competing interests exist.

## Introduction

The spleen plays an important role in protection against infection and individuals with loss of spleen function are at an increased risk of infection with encapsulated microorganisms and systemic spread [1]. In sickle cell disease (SCD), the spleen undergoes a sequence of pathological changes that ultimately result in loss of splenic function [2], and this accounts for most of the early morbidity and mortality associated with this disorder [3]. The tests often used to assess spleen function are based on scintigraphy, haematological, and immunological techniques [4, 5]. The scintigraphy method is the gold standard and assesses the ability of the spleen to filter blood of abnormal cells and particles. However, this method is expensive and involves injecting radio-labelled substances into patients [5–8]. The haematological methods reflect the inability of the spleen to phagocytose poorly deformable red blood cells (RBCs) or those containing inclusions. Splenic dysfunction is usually evaluated on the basis of increased numbers of such abnormal RBCs in circulation. The presence of Howell-Jolly bodies (HJB) is one such example; percentages of HJB red cells can be estimated from peripheral blood smears using the classical May-Grunwald Giemsa stain (MGG) [5, 9], by flow cytometry or imaging flow cytometry methods [10–13]. In patients with diminished splenic function, the membrane of the red blood cells contains surface indentations referred to as 'pits' when viewed using contrast-enhancing microscopy techniques such as interference phase-contrast [3, 14, 15] and differential interference contrast (DIC) microscopy [5, 16, 17]. Recently, a new fluorescence-based method and an automated deep neural network for quantifying pitted red cell counts have been developed [18, 19]. The pitted red cells count method represents a sensitive technique for evaluating splenic function; however, it requires specialised equipment and personnel. Another method of evaluating spleen function is counting red cells containing silver-stained (argyrophilic) inclusions (AI). The silver stain was originally developed to study the nucleolar organizer regions (NORs) of chromosomes and to evaluate their function [20, 21]. In addition to staining the argyrophilic proteins associated with NORs, the stain can detect non-NOR silver-staining cytoplasmic granules and inclusions such as hemosiderin (appear as dark-brown to black), calcium and phosphate when applied to marrow smears [22, 23]. The silver stain technique was first employed to assess splenic function by a group of investigators in the USA, following their observation that red cells from SCD patients and splenectomised individuals contained large numbers of AI [24]. The AI count method is non-invasive and requires only a light microscope.

Spleen function has not been well studied in SCD patients residing in Africa [25], the region with more than two thirds of the global burden of SCD [26]. This is mainly because most of the methods mentioned above are not readily available in most low-income settings of Africa as evidenced by the scarcity of data in the literature [14, 27]. The HJB and AI methods may be suitable but have not been widely used in Africa; both tests require only a light microscope and therefore are feasible in Nigeria and most low-to-middle-income countries. They can be used to facilitate the measurement of splenic function in SCD patients where resources for spleen scintigraphy and interference contrast microscopy are absent. Improvement in spleen function has been reported following chronic hyper-transfusion [28, 29] and following hydroxyurea therapy; the splenic filtration function was preserved after three years in a third of the patients following hydroxyurea therapy [30, 31]. Therefore, assessment of splenic function among SCD patients in low-income countries can play a vital role in identifying those patients who can benefit from such therapy. The objective of the current study was to assess splenic function by comparing the frequencies of red cells containing HJB and AI in SCD patients with those of normal controls. We also evaluated which of the method was associated with the least intra-observer variation. If splenic function could be assessed in this manner, these techniques

might be utilized for early identification of SCD patients at risk of developing severe infection because of reduced or absent splenic function, who may benefit from more intensive infection prevention measures.

## Methods

### Study participants

This was a prospective cross-sectional study conducted amongst children and adults with SCD in steady state between September 2020 and November 2021; all patients who presented to the outpatient paediatric and adult haematology clinics of the University of Maiduguri Teaching Hospital (UMTH) during the study period were invited to participate. We excluded patients who had received blood transfusion in the last 3 months as this could interfere with the peripheral blood counting for AI and HJB red cells, as well as those with other acute or chronic illnesses that might interfere with spleen function. Apparently-healthy individuals consisting of medical students and children of hospital staff, with no recent history of fever or evidence of any underlying haematological disorder were enrolled as controls. The controls were of similar age range to the SCD participants. All study participants underwent abdominal ultrasonography to document the presence or absence of the spleen. Based on spleen length data obtained from the control group, spleen sizes among the SCD patients were classified into three categories as—small, normal, and enlarged as previously described [32].

### Ethics

Institutional review board approval was obtained from both UMTH (REC reference number: 20/606) and Liverpool School of Tropical Medicine (REC reference number: 20–010). Written and verbal consent was obtained from all the participants.

### Laboratory analysis

**Blood smears.** Two thin blood smears, each for AI and HJB red cells estimation were made from blood samples collected by venepuncture from the study participants within 4 hours of collection and allowed to air dry. Smears for AI red cells were fixed for 3 minutes in a solution made by diluting 150 ml ethanol to 450 ml formalin (37% formaldehyde), washed in distilled water, and allowed to dry thoroughly before proceeding to staining. Smears for HJB red cells were fixed in absolute methanol for 1 minute and allowed to air dry before staining with May-Grunwald Giemsa (MGG) as described below. All fixed smears were stored in slide boxes if staining was not performed immediately.

### Method for AI count using silver stain

The silver stain method was based on the technique described previously [24], with some modifications (S1 Text). A pilot study to assess and refine the test performance was performed using samples from controls and SCD patients. We conducted a pre-test run to check the effect of staining duration, temperature, the constitution of eosin, and pre-treatment of slides with potassium iodide on optimal staining conditions. As a result, the staining duration was reduced to 20 minutes in the final study protocol, as a longer duration was associated with over-stained slides that could not be interpreted (S1 Fig). The eosin (Thermo Fisher scientific) was constituted using an aqueous solution as this provided better counterstaining ability, compared to an alcohol-based formulation. The reduction of slides with potassium iodide did not produce any added effect; therefore, this step was not included in the final protocol. Furthermore, two different brands of silver nitrate salts (50%; Honeywell Fluka and FujiFilm) and

Gelatin powder (500g: Sigma-Aldrich and Riedel De Haen) were tested and both gave the same staining quality.

Finally, not more than 3 ml of working solution was prepared per staining session, as the stain began to deteriorate after about 3 minutes of preparation (S2 Fig). The slides were stained using silver stain at 38˚C in dark for 20 minutes as described (S1 Text). All stained slides were mounted with coverslips (22 x 40 mm; thickness 0.13 to 0.16 mm) using a permanent mount (DPX); this improved the clarity of microscopy and reduced the effect of reflection observed with unmounted slides.

## Method for quantitative HJB count using May-Gruenwald Giemsa (MGG) stain

To obtain a uniform quality of staining, the slides were stained simultaneously in batches using a staining tank containing freshly made May-Gruenwald (Sigma-Aldrich) working solution for 5 minutes and washed in 2 to 3 changes of distilled water. The slides were next immersed in Giemsa stain (Lab Tech Chemicals) for 10 minutes before washing in 2 to 3 changes of distilled water. The slides were allowed to dry upright as this prevented residue from forming on the slides. Any residual water on the slides was wiped off with dry gauze, which also improved the clarity of the film. The HJB slides were not mounted as this did not affect the clarity of the slide.

## Microscopy

Microscopy was performed with an upright light microscope (Leica DM 750, equipped with a ICC50 E colour camera). Smears were examined using the x40/0.65 and x100/0.80 (oil immersion) HI PLAN objectives. For the AI count, a minimum of 500 consecutive RBCs per smear were examined for the presence of one or more distinct black granules. Red cells with diffuse or fine reticular pattern of brown staining were not regarded as positive [24]. For the HJB count, 400 RBCs were counted per smear for the presence of a distinct purple to dark-purple dot. Red cells with more than one dots were not counted. The AI and HJB counts were expressed as percentages of the total red cells counted. Two blood smears each for AI and HJB red cells from each study participant when available were examined separately for the percentages of cells of interest, the mean of the two counts observed was then calculated. All counts were performed directly in the microscope eyepiece. To remove intra-observer variation and improve the precision of the microscopy results, all counts were performed by a single person (AIL). The reader had undergone prior training and quality checks during the pilot phase of the project.

## Statistical analysis

The data were entered into Excel spread sheet for cleaning and sorting and exported into Statistical Package for the Social Sciences (SPSS) (version 25; SPSS, Chicago, IL, USA) and MedCalc v.20.114 (MedCalc Statistical Software Ltd, Ostend, Belgium) for analysis. Categorical data were summarized using frequency and proportions while continuous data were expressed as means, median and interquartile range as appropriate. For the AI and HJB methods, the intra-observer reliability and agreement for paired measurements performed for an individual sample were assessed. Firstly, scatter plots were drawn to assess the relationship between the two separate readings. The Pearson's correlation analysis was used to assess the intra-observer reliability for the paired readings obtained using each method. The Bland Altman analysis was used to measure the limit of agreement between the two separate readings [33]; individual values were expressed relative to the geometric mean of the first and second counts for that

sample. A non-parametric test was used to compare results between SCD patients and controls as appropriate. The level of significance was set at the two-tailed P-value <0.05.

## Results

### General characteristics

Blood smears for identification of red cells containing AI and HJB were obtained from 182 SCD patients (175 Hb SS, 5 Hb SC, and 2 Hb SB thal) and 102 controls (93 Hb AA, 7 Hb AS, and 2 Hb AC). The median ages for the SCD group and controls were 11.0 years (range 1–45 years) and 12.0 years (range 1–32 years) respectively.

### Identification of argyrophilic inclusion-positive (AI) and HJB red cells

Both AI- and HJB-containing red cells could be easily identified in the participants; representative smears for AI and HJB are shown below; the AI inclusions occurred either as single or multiple inclusions within a red cell (Fig 1). The HJB appeared as single, spherical inclusion often situated close to the cell periphery (Fig 2).

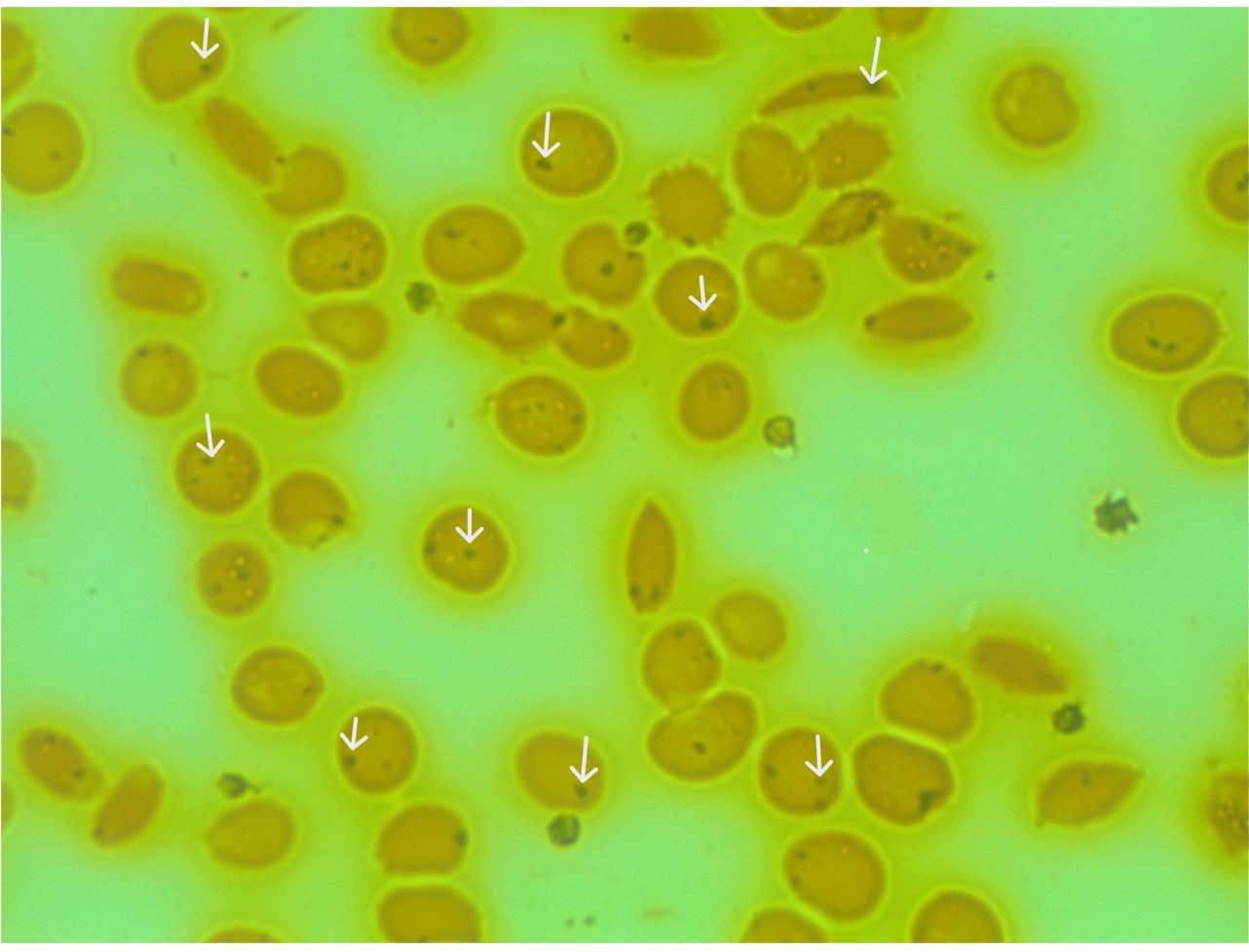

**Fig 1. Detection of argyrophilic inclusion (AI) containing red cells.** Silver-stained blood smear (x100 / 0.80) from a 16-year-old male HbSS patient. Several red cells contain black granules corresponding to the argyrophilic inclusions (white arrows). The inclusions varied in size and numbers. Several sickled red cells also noted.

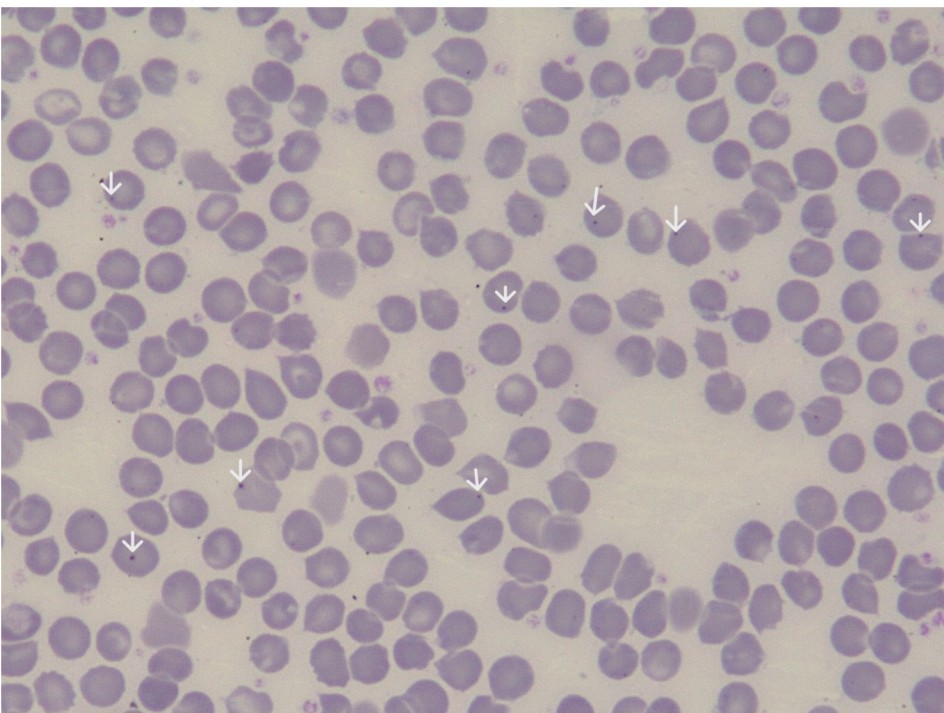

**Fig 2. Detection of Howell Jolly body (HJB) containing red cells.** May-Grunwald Giemsa (MGG) stained blood smear (x40 / 0.65) from a 23-year-old female HbSS patient showing several red cells with HJB (white arrows).

The distribution of HJB and AI red cells across the study population is shown in Table 1. We compared the range of results obtained for red cell inclusions using both the AI and HJB methods. It was observed that the results were always higher with the silver stain method for AI red cells. Among the SCD group, the AI red cell counts ranged from 34.5% to 66.0% and HJB counts ranged from 0.7% to 3.1%. Among the controls, the AI red cells count ranged from 5.1% to 8.7% whereas the HJB counts ranged from 0.1% to 0.5%.

**Table 1. Distribution of AI and HJB red cell counts across study population.**

| Hb phenotype | N (%) | Sex (m) n, % | % HJB RBC | | % AI RBC | |
|---|---|---|---|---|---|---|
| | | | Median(IQR) | Min -Max | Median (IQR) | Min-Max |
| **SCD group** | | | | | | |
| Hb SS | 175 (95.9) | 87 (49.7) | 1.4 (0.8–3.1) | 0.0–28.2 | 46.3 (35.0–66.0) | 3.7–89.0 |
| Hb SC | 5 (2.9) | 4 (80.0) | 2.2 (1.2–3.1) | 0.3–3.7 | 62.5 (27.0–74.0) | 2.0–74.2 |
| Hb SB Thal | 2 (1.1) | 2 (100.0) | 1.2 (0.7–*) | 0.7–1.8 | 35.2 (16.0–*) | 16.0–54.5 |
| **All** | **182 (100)** | **93 (51.1)** | **1.5 (0.7–3.1)** | **0.0–28.2** | **47.4 (34.5–66.0)** | **2.0–88.9** |
| **Control group** | | | | | | |
| Hb AA | 93 (91.1) | 56 (60.2) | 0.3 (0.1–0.5) | 0.0–1.2 | 6.9 (5.1–8.6) | 2.0–15.3 |
| Hb AS | 7 (6.9) | 6 (85.7) | 0.3 (0.0–0.9) | 0.0–2.7 | 7.5 (4.0–13.1) | 3.7–13.7 |
| Hb AC | 2 (2.0) | 0 (0.0) | 0.4 (0.2–*) | 0.2–0.5 | 8.9 (7.6–*) | 7.6–10.2 |
| **All** | **102 (100)** | **62 (60.8)** | **0.3 (0.1–0.5)** | **0.0–2.7** | **7.1 (5.1–8.7)** | **2.0–15.3** |

AI: Argyrophilic inclusions; HJB: Howell Jolly Bodies; RBC red blood cells; Hb: haemoglobin.

* 75th centile is not generated because of only 2 data in the group

## Comparison of AI and HJB red cells count among the study population

The median percentage of AI red cells in the SCD group (47.4%; IQR 34.5%–66.0%) was significantly higher than those of the control group (7.1%; IQR 5.1%–8.7%) ($P < 0.0001$) (Fig 3A). Similarly, the median percentage of HJB red cells in the SCD group (1.5%; IQR 0.7%–3.1%) was greater than those of the control group (0.3%; IQR 0.1%–0.5%) ($P < 0.0001$) (Fig 3B). Analysing the results according to Hb phenotype (Kruskal- Wallis test), within the SCD group, the HJB and AI red cell counts tended to be higher in the HbSS population compared to the other compound heterozygotes, however, the results were not statistically significant either for the HJB ($P = 0.883$) or AI counts ($P = 0.534$). The small number of the latter group (5 Hb SC and 2 Hb SB Thal) may have affected the power to detect any statistical significance among the SCD group. Among the control group, the median percentage of AI red cells for those with Hb AA was not significantly different from those with Hb AS and Hb AC ($P = 0.296$). Similarly, the frequency of HJB red cells in the Hb AA controls was comparable to those of Hb AS and Hb AC controls ($P = 0.946$).

## Comparison of frequency of AI and HJB red cells based on spleen size assessed on ultrasonography

The SCD patients were divided into two groups based on whether the spleen was visible (n = 99) or not visible (n = 73) on abdominal ultrasonography. The median percentage of AI red cells was significantly higher in the group without visible spleens (56.0%; IQR 38%–73%) compared to those whose spleens were visible on ultrasonography (44.9%; IQR 29%–61%) ($P = 0.011$) (Fig 4A). Similarly, the median percentage of HJB red cells in the group without visible spleens on ultrasonography was higher (2.2%; IQR 1.5%–5.1%) compared to those whose spleens were visible (0.9%; IQR 0.6%–2.0%) ($P = 0.0001$) (Fig 4B). Furthermore, among the SCD patients with visible spleens, the spleens were classified as small (n = 18), normal (n = 62), and enlarged (n = 19) using spleen length of the controls as reference. The frequencies of AI and HJB red cells were compared across subgroup of spleen sizes. The proportion of AI red cells showed an inverse relationship with spleen size among the whole study cohort (Fig 4C). The proportion of HJB red cells was however not different among SCD patients with small, normal, or enlarged spleens (Fig 4D).

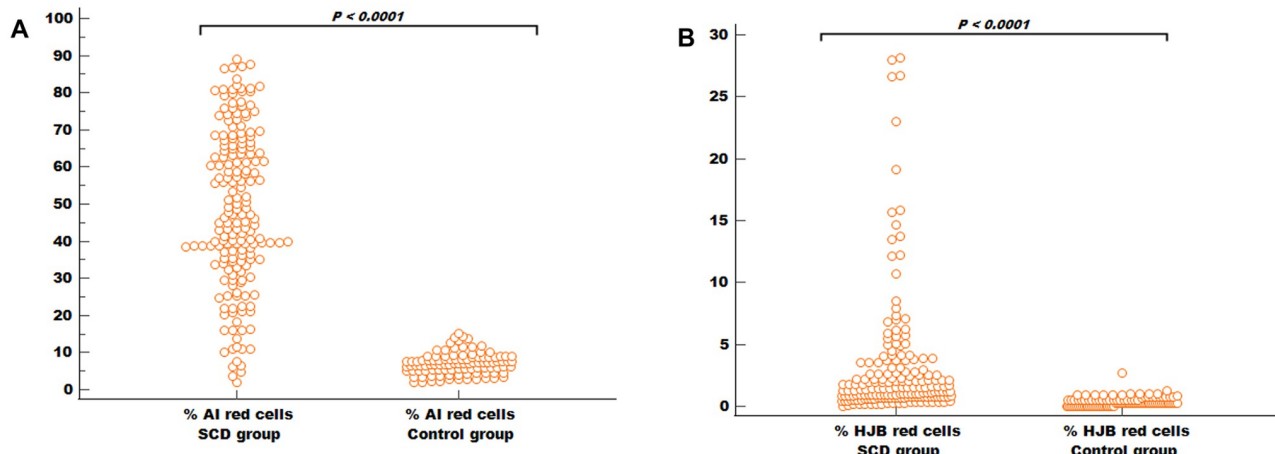

**Fig 3. Comparison of AI and HJB red cells count among the study participants.** Plots showing significantly high proportion of AI red cells (A) and HJB (B) red cells among the SCD patients (n = 182) compared to the control group (n = 102). Mann-Whitney U test was used (***: P < 0.0001).

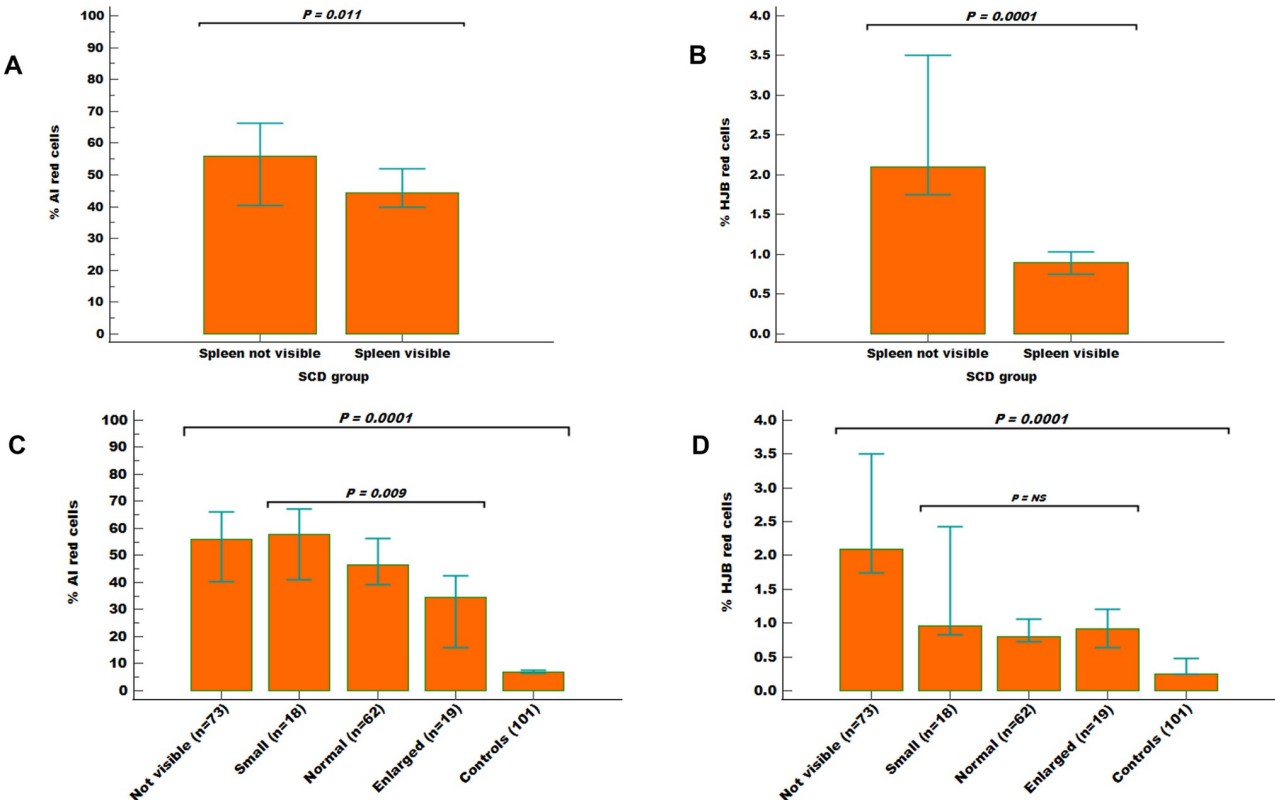

**Fig 4. Comparison of AI and HJB red cells among the study participants based on spleen size.** Bar charts showing significant difference in the proportion of AI (A) and HJB (B) red cells between SCD patients with visible spleens (n = 99) and those without visible spleens (n = 73) on ultrasonography. Mann-Whitney U test was used. Further categorization of AI and HJB red cell counts based on different spleen sizes among the SCD group (i.e not visible, small, normal, and enlarged) and the controls was made (C and D). The proportion of AI red cells (C) showed an inverse relationship with spleen sizes among the SCD (P = 0.009). The proportion of HJB red cells (D) was not significantly across the different spleen sizes. Error bars represent the 95% Confidence interval for median. Kruskal-Wallis and Dunn's multiple comparison tests were used for the analysis between the SCD and control group.

## Intra-observer reliability and agreement between counts obtained for the AI and HJB method

Scatter plots for paired readings obtained from the same sample for both the AI and HJB methods are shown below (Fig 5A to 5D). The Pearson's correlation (r) for the AI and HJB counts was 0.90 ($r^2$ = 0.82; P <0.001) and 0.92 ($r^2$ = 0.86; P <0.001) respectively. Plots of the paired measurements of the AI and HJB counts on the original scale (Fig 5A and 5C respectively) shows a cluster of points around the lower part of the graphs and a few outlying values. Use of log scales for the AI and HJB counts (Fig 5B and 5D respectively) results in a more uniform scattering of the points on the graph and a scatter that was roughly constant with increasing count. The corresponding correlation coefficient (r) was 0.86 for the AI and 0.70 for the HJB red cell counts.

## Intra-observer agreement between counts obtained for the AI and HJB method

The Bland-Altman test was used to evaluate the intra-observer agreement for the two methods. The results of the AI count showed a weak agreement between the two sets of counts when all the data sets were analysed (i.e., both SCD and control data) (Fig 6A). The mean difference

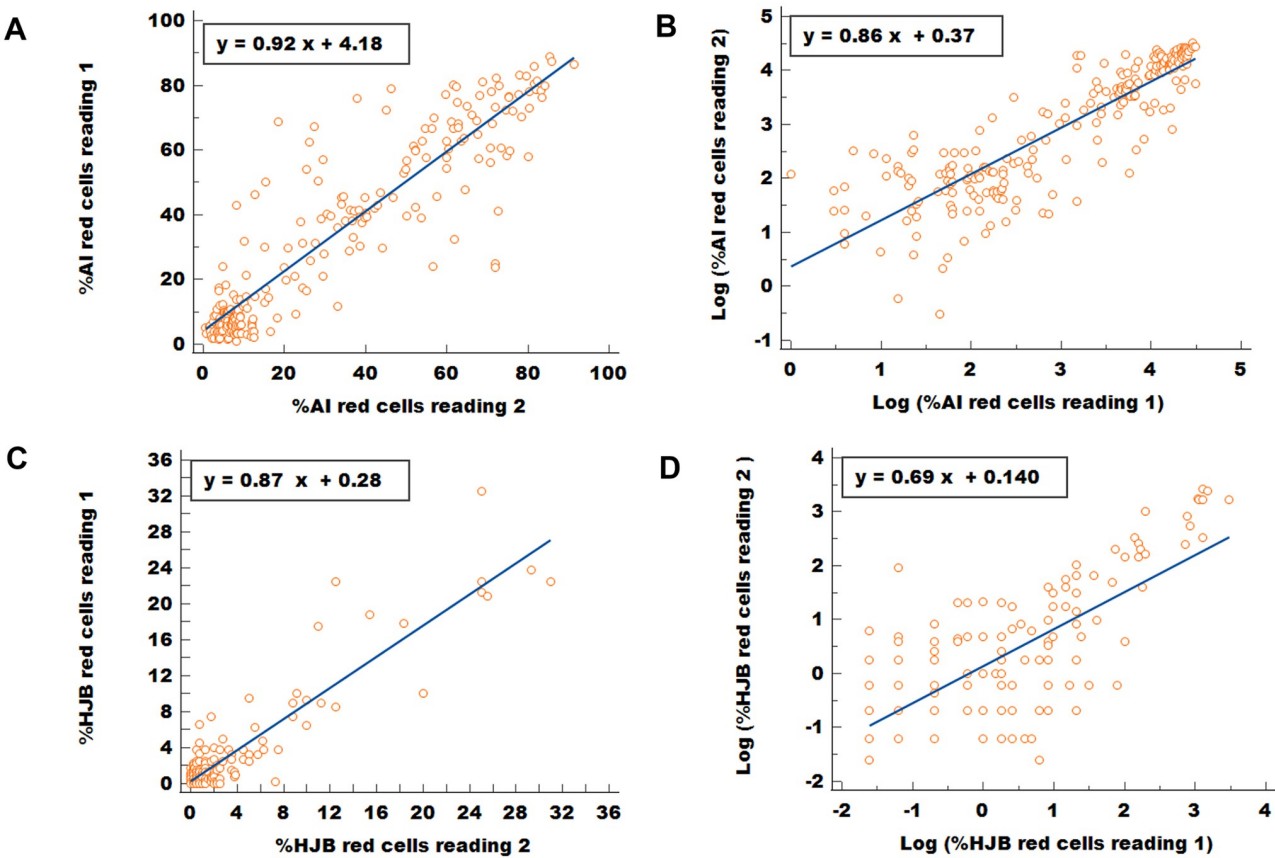

**Fig 5. Graphs showing intra-observer reliability for the AI and HJB counts.** A) Scatter plot for paired measurements for the AI red cell count (y = 0.924x + 4.18) (r = 0.90). A total of 233 paired samples (both SCD and controls) were analysed. B) Plot showing the log transformed AI red cell count from A above (y = 0.858x + 0.370) (r = 0.86). C) Scatter plot for paired measurements obtained for the HJB red cell counts (y = 0.867x + 0.277) (r = 0.92). A total of 174 paired samples (SCD patients only) were analysed. D) Plots for the log transformed paired measurements of HJB red cells count from C above (n = 138; HJB values zero or below were not log transformed and were excluded (y = 0.687x + 0.140) (r = 0.70).

between the two sets of readings was high (difference 1.7%; 95% Confidence interval (CI): 0.2% to 3.3%) (*P* = 0.03). The upper limit of agreement (LOA) was 25.6% (95% CI: 22.9% to 28.3%) and the lower LOA -22.1% (95% CI: -24.8% to -19.4% (Fig 6A). Nineteen measurements (8.2%) fell outside the LOAs, all were from samples with AI count of > 20%. However, when only the data sets from the control population were analysed, there was good agreement between the two sets of reading evidenced by the significantly low mean difference of 0.35 (95% CI: -0.7% to 1.4%) (*P* = 0.526) (Fig 6B). The upper LOA was 10.6% (95% CI: 8.8% to 12.5%) and the lower LOA -9.9% (95% CI: -11.8% to -8.1%). The mean difference between the two sets of HJB readings was low (difference -0.1%; 95% CI: -0.4% to 0.2%) (*P* = 0.579) (Fig 6C). Although, the points appeared clustered towards the lower end, they were scattered relatively evenly above and below the mean line of equality. The upper limit of agreement (LOA) was 4.3% (95% CI: 3.7% to 4.9%) and the lower LOA -4.5% (95% CI: -5.1% to -3.9%. Eleven measurements (6.3%) fell out of the LOAs, and majority were from samples with a HJB count of >5%.

## Discussion

Spleen function is usually evaluated by the spleen's ability to remove abnormal cells from circulation. SCD patients develop progressive splenic dysfunction and quantification of red cells

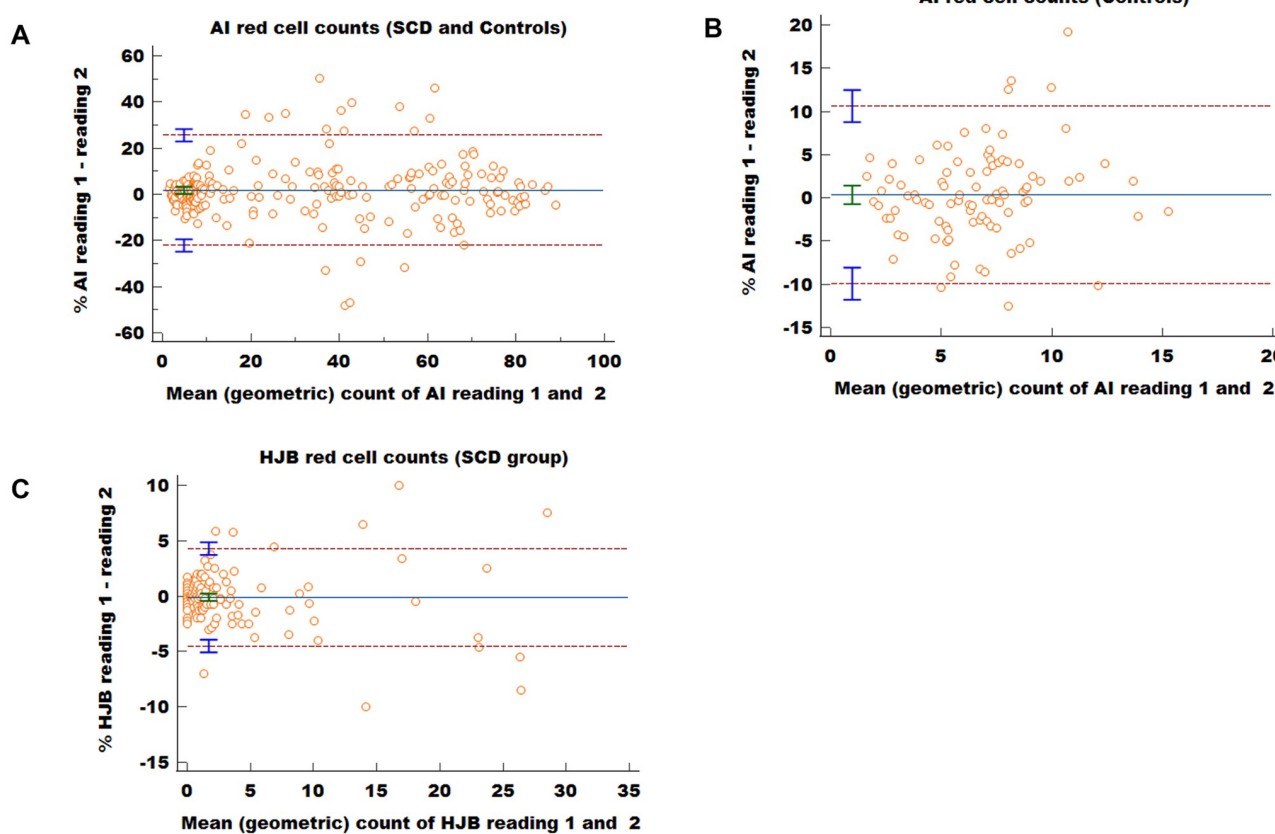

**Fig 6. Bland Altman (BA) plot comparing paired measurements obtained for both the HJB and AI counts.** (A) BA plot of paired AI measurements in 233 SCD and control subjects. Mean difference between measurements = 1.7% (95% CI: 0.17% to 3.3%, horizontal middle line). Upper limit of agreement = 25.6% (95% CI: 22.9% to 28.3% upper dashed line). Lower limit of agreement = −22.1% (95% CI: −24.8% to −19.4%, lower dashed line). (B) BA plot of paired AI measurements in 93 controls; mean difference between measurements = 0.35% (95% CI: -0.7% to 1.4%, horizontal middle line). The uniform distribution of the points around the mean suggest they are in agreement. Upper limit of agreement = 10.6% (95% CI: 8.8% to 12.5%, upper dashed line). Lower limit of agreement = −9.9% (95% CI: −11.8% to −8.1%, lower dashed line). (C) BA plot of 174 paired measurements for HJB count from SCD patients only. Mean difference between measurements = -0.1% (95% CI: −0.4% to 0.2%, horizontal middle line). Majority of the points centred around the mean and were clustered towards the lower end; there was an increasing variability with higher counts to the right. Upper limit of agreement = 4.3% (95% CI: 3.7% to 4.9%, upper dashed line). Lower limit of agreement = −4.5% (95% CI: −5.1% to −3.9%, lower dashed line).

containing two types of inclusions—HJB and AI, was employed in the current study to evaluate their splenic function. The silver stain method used in the current study to evaluate AI-containing red cells as a marker of spleen function was based on that first described by Tham et al. [24]. The exact nature of the AI inclusions is unclear; however, studies have demonstrated the presence of vacuoles and inclusions among circulating red cells [34, 35]. The inclusions were increased in individuals without spleens or in those with abnormalities of erythropoiesis. They were more abundant in splenectomised persons with concurrent hematologic disorders such as thalassemia and haemolytic anaemia. Their increased presence in abnormalities of erythropoiesis may be readily explained by the larger amount of product of haemolysis to be disposed of in these conditions. Their increase however in splenectomised individuals, with otherwise normal erythropoiesis, suggests an intact spleen is required for their elimination. In one study, the authors noted inclusions in 54.3% of red cells from 20 splenectomised individuals, using differential interference contrast microscopy [35]. These inclusions are probably equivalent to the AI inclusions observed in the current study using the silver stain, in close

agreement with our observed mean value of about 50% for the AI red cells. The authors noted the inclusions neither stained nor looked like conventional red cell inclusions, such as HJB, Heinz bodies or siderotic bodies. Morphologically, the inclusions were suggestive of haemoglobin degeneration. They concluded that mature normal red cells continually form inclusions which may reflect the degradative cellular process consequent upon cell aging. Kent and colleagues also observed, by electron microscopy, the presence of autophagic vacuoles in mature red cells and reticulocytes; the inclusions contained a variety of materials including haemoglobin and altered cytoplasmic organelles such as ribosomes, mitochondria, and smooth membranes [34]. The inclusions appeared to be instrumental in the disposal of those organelles or other materials not required by the fully mature red cells. The increased presence of these inclusions in splenectomised subjects indicates that, although developing human red cells are capable of eliminating their digestive residues, the presence of the spleen is required to fulfil this task adequately.

The use of MGG to demonstrate HJB in red cells is a long-established method of evaluating splenic dysfunction [3, 36, 37]. HJB are small (approximately 1 micron) red cell DNA inclusions that result from cytogenetic damage. Normally, HJB-containing red cells are formed at low frequency and are quickly removed by an intact spleen, thus their presence in peripheral smears is an indirect evidence of splenic dysfunction [37, 38]. Our comparison of results obtained using HJB and AI inclusions as markers of splenic dysfunction showed the AI in red cells were always higher than the HJB counts. This indicates the two tests identify different intra-cellular structures. While the silver stain picks up all intracellular argyrophilic particles, MGG stains only HJB, which are usually formed at low frequency. The argyrophilic inclusions varied in size and number within the red cells, whereas the HJB always appeared as a single inclusion. The higher AI count may also be related to the increased erythropoiesis seen in haemolytic disorders like SCD. Kent et al. found increased red cells with inclusions in patients with haematological disorders and reticulocytosis but intact spleens; a high proportion of the reticulocytes contained inclusions [34]. Recent studies have demonstrated significantly high fraction of RBCs retaining-mitochondria in circulation among SCD patients; [39, 40]; the abnormal presence of mitochondria was thought to be related to the stress erythropoiesis observed in SCD. Aside its utility in assessing argyrophilic nucleolar organising regions, the silver stain can stain other elements including iron, calcium, and phosphate [22]. Application of the silver stain to a series of marrow particle sections showed the silver stain was specific for identifying abnormal mitochondrial "iron" deposits in ringed sideroblast [22]. Given the heterogenous nature of red cell vacuoles described above [34, 35], the sensitive but non-specific nature of the silver stain may allow it to stain a wide range of cellular content. Whatever their nature, it appears the AI are found in increased numbers in SCD patients compared to HJB. The forma, although more sensitive in picking up red cells inclusions, may be influenced by both splenic dysfunction and haemolysis related abnormalities.

## Comparison between SCD patients and controls

To our knowledge, this is the first detailed study of splenic function among SCD patients in an African setting. Our study found significant differences in the frequencies of AI and HJB red cells between the SCD patients and controls, which indicates the presence of splenic dysfunction in the former. The percentage of AI red cells in our SCD population was significantly higher than those of the controls; such a finding using the AI red cells has been reported in the US [24], though the study only included 9 SCD patients and 45 controls. The results obtained for the AI red cells in our SCD patients (IQR 34.5%–66.0%) and controls (IQR 5.1%–8.7%) were higher than the data from the SCD patients (11.8%–52.7%) and controls (0.3%–3.0%) in

the American study [24]. The SCD patients in the present study, demonstrated a higher frequency of HJB red cells compared to the controls. The range of HJB red cells obtained among our SCD population was higher (IQR 0.7%–3.1%) than results obtained among SCD patients (n = 12; age range 5 months to 39 years) in the United States (HJB% range 0.0%–1.1%) [17], and among children (n = 20; age range 5 years to 22 years) with SCD in Brazil (HJB% range 0.0%–1.4%) [41]. The small sample size from both studies compared to our sample population (n = 182) may account for the differences compared to our results.

## Comparison of AI and HJB red cells with spleen size on ultrasonography among the study participants

Both the AI and HJB red cell counts from SCD patients without visible spleens (i.e autosplenectomy) were higher than patients whose spleens were visualized on ultrasonography, especially for the HJB red cell counts. This is not unexpected as the spleen is the site of removal of red cell inclusions, therefore patients with autosplenectomy are likely to have higher numbers of inclusions within their red cells than those with intact spleens [12, 18]. However, despite the presence of spleens among some of the SCD patients, the frequency of red cell inclusions was still higher than in the controls. This suggests the presence of functional hyposplenism among SCD patients, whereby despite having the spleen present, it may not be effectively eliminating the inclusions [38]. Previous studies have shown that function can be restored among such group of patients using transfusion or disease modifying agents like hydroxyurea [28, 29, 31]. Furthermore, among SCD patients whose spleens were still visible on ultrasonography, the AI and HJB counts appeared variable among those with small, normal, and enlarged spleens, with a tendency of been lower in those with enlarged spleens. The implication of this finding is that the spleen function may not be completely lost. A recent study demonstrated that spleen function may be preserved even among adults patients with SCD as long sinus structures still persist in the preserved or partially damaged spleen [18]. It would be interesting to follow up the group of patients whose spleens are still visible on ultrasonography with disease modifying therapy like hydroxyurea to determine if this will have any effect on reducing the frequency of the red cell inclusions.

## Reliability and agreement analysis for the HJB and AI methods

To evaluate which of the method would give the least intra-observer variation and be useful for the longitudinal follow up of our patients, we evaluated the intra-observer reliability and agreement of both methods. Our data demonstrated good intra-observer correlation for paired readings obtained with the AI test (r = 0.90; $r^2$ = 0.82), in keeping with an earlier report (r = 0.736; $P$ <0.001) [24]. While correlation measures the linear relationship between two sets of measurement, agreement on the other hand assesses the equality of the individual values between two sets of measurement [42]. Since a perfect positive correlation is not proof of equal responses between two testing occasions [43], we tested results obtained for both the AI and HJB methods for agreement using the Bland Altman analysis [33]. A good agreement was observed for the AI counts among the controls ($P$ = 0.499), but at higher counts obtained among the SCD patients, the AI method did not produce such a good agreement between two readings taken on different occasions. This finding reflects the character of the AI test; the difference between two separate readings taken on the same sample is related to the value of the measurement, larger measurements among the SCD patients imply a larger average error between both readings and hence the significant difference between these counts ($P$ = 0.03).

Our data also showed a good limit of agreement for counts obtained using the HJB method (95% CI, -4.5% to 4.3%; $P$ = 0.579). Several investigators have shown the HJB count can be

reliably used to assess spleen function. Corazza et al found a significant correlation between HJB red cell counts obtained by the classical MGG and the pitted red cells counts (P < 0.0001); they noted that pitted red cells count above 8% was always associated with increasing HJB red cells count [9]. Serial measurements of HJB red cells using the manual estimation were reliably used to monitor spleen function before and after bone marrow transplantation in SCD patients over a period of 15 years [44, 45]. In a similar, but small study, the frequency of HJB red cells declined progressively following bone marrow transplantation, and together with scintigraphy, both techniques were used to monitor spleen function in these patients [46]. Also, a robust correlation between the MGG method and a newly-developed imaging flow cytometry method for evaluating HJB has been demonstrated [10, 13]. A few studies however have reported a lack of correlation between the MGG HJB method with other methods of assessing spleen function [5, 17, 41].

## Clinical implication

A prospective study of SCD patients (n = 12) identified at birth in a screening programme in the USA, demonstrated that splenic dysfunction is an acquired defect occurring as early as five months of age; the onset of splenic dysfunction documented by serial splenic scintigraphy correlated with the appearance of HJB in peripheral smears [36]. This suggests that the presence of HJB can be used as a reliable indicator of the onset of splenic dysfunction. Of note however, none of the studies that have evaluated HJB manually as a marker of splenic dysfunction have indicated what percentage of red cells containing HJB indicates hyposplenism. Using flow cytometry, quantitative HJB values of $>300/10^6$ red cells (i.e 0.03%) have been suggested to indicate loss of splenic filtration among SCD [11]. We have included a control group in our study and our result for the percentage of HJB among the controls obtained with the manual count (median 0.3%; IQR 0.1%–0.5%) is comparable to that obtained from children and adult healthy controls (0.3%; range, 0.01%-0.6%) in studies using flow cytometry counts of HJB [13, 47]. This indicate that the measurement of HJB by the MGG technique using light microscopy may allow accurate evaluation of splenic function among SCD patients in low resource setting.

The 97.5[th] centile (non-parametric upper reference limit) for HJB counts (0.9%) generated from our controls could be considered as upper limit of normal and values above this may be used to indicate presence of splenic dysfunction. However, for the longitudinal follow-up of patients, simple comparison to a particular threshold value may be overly simplistic, because splenic dysfunction among SCD is known to be progressive [3, 48], and there is great inter-individual variability in the HJB counts among children and adults with SCD [11, 13, 37]. Rather, identifying a baseline value for an individual patient may aid the early detection of splenic dysfunction when a considerable increase from the baseline HJB level occur [37, 47]. Also, while some studies have used the appearance of HJB to identity the onset of splenic dysfunction [3, 36], other studies have used disappearance or decreasing levels of HJB to monitor response to therapy such as bone marrow transplant and hydroxyurea [31, 45, 46] since it indicates the return of splenic function among SCD patients. Therefore, in clinical practice serial monitoring of an individual's HJB level could help in monitoring spleen function and guide the appropriate timing of interventions rather than comparison to a threshold. Compared to other currently available methods of assessing spleen function, such as spleen scintigraphy, pitted red cell counts and flow cytometry, counting of HJB red cells by the MGG method provides a simple and reliable technique that can easily be used in most laboratories in the African settings. HJB was easy to perform, simpler than the AI count method and can be used to evaluate splenic function at regular intervals. This will be valuable in identifying early the onset of splenic dysfunction, which is associated with increased susceptibility to overwhelming infections.

## Limitations

To determine estimates of diagnostic accuracy such as sensitivity and specificity, we would have compared our findings with those of a reference standard such as the spleen scintigraphy or pitted red cell count [49]. Since these methods are not available in Nigeria, we have compared our findings to those in the published literature.

## Conclusions

Determination of red cells with AI and HJB enables the assessment of splenic function in SCD patients in low-resource settings. Both markers can serve as indicators of splenic dysfunction in SCD patients as reflected by the high percentages of circulating levels compared to controls. Their presence was easily demonstrable, and both showed good intra-observer reliability. Although the red cells inclusions were readily demonstrable using the AI method, the silver stain deteriorates rapidly. The HJB method is simpler and requires fewer reagents than the AI method. Moreover, the higher AI counts in patients with haemolytic anaemias such as SCD may be due to both the hematologic disturbance and abnormal splenic function, therefore, further validation in larger studies may be required for the AI method before the generalisability of its efficacy in assessing spleen function in SCD can be ascertained.

## Supporting information

**S1 Fig. Effect of prolonged staining of slide.** Silver-stained blood smear (x100/0.80) from a 22-year-old female HbSS patient showing deeply stained red cells (white arrows) making visualization of some of the argyrophilic inclusions difficult.
(TIF)

**S2 Fig. Silver stain reaction time.** A) Silver stain at 1 minute. The preparation appears clear and colourless. B) The preparation has begun to take a yellow colouration. C) The preparation has turned golden yellow and continues deteriorating on further standing.
(TIF)

**S1 Text. Standard operating procedure for silver stain technique.**
(DOCX)

**S1 Data.**
(ZIP)

## Acknowledgments

The authors are grateful for the assistance of the staff of Histopathology and Haematology department of UMTH during data collection, sample preparation and microscopy analysis of this project.

## Author Contributions

**Conceptualization:** Adama I. Ladu, Imelda Bates.

**Formal analysis:** Adama I. Ladu.

**Investigation:** Adama I. Ladu, Ngamarju A. Satumari.

**Methodology:** Adama I. Ladu, Ngamarju A. Satumari, Aisha M. Abba, Fatima A. Abulfathi.

**Supervision:** Caroline Jeffery, Adekunle Adekile, Imelda Bates.

**Writing – original draft:** Adama I. Ladu.

**Writing – review & editing:** Adama I. Ladu, Ngamarju A. Satumari, Aisha M. Abba, Fatima A. Abulfathi, Caroline Jeffery, Adekunle Adekile, Imelda Bates.

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
