## [Decision Letter · Decision Letter 0]

2 Mar 2023

PGPH-D-22-02120

Evaluation of two red cell inclusion staining methods for assessing spleen function among sickle cell disease patients in North-East Nigeria

Dear Dr. Ladu,

Thank you for submitting your manuscript to PLOS Global Public Health. After careful consideration, we feel that it has merit but does not fully meet PLOS Global Public Health’s publication criteria as it currently stands. Therefore, we invite you to submit a revised version of the manuscript that addresses the points raised during the review process.

The manuscript has been evaluated by three reviewers, who provided a comprehensive assessment of the study and its underlying methodology. The reviewers suggested further discussion regarding potential confounding variables, clinical dimensions, and AI compositions. Interesting methodological considerations have also been raised which require clarification.<o:p></o:p>

Could you please carefully revise the manuscript to address all comments raised?<o:p></o:p>

We look forward to receiving your revised manuscript.

Kind regards,

Lucinda Shen, MSc

Staff Editor

Journal Requirements:

Additional Editor Comments (if provided):

Reviewers' comments:

Reviewer's Responses to Questions

**Comments to the Author**

1. Does this manuscript meet PLOS Global Public Health’s publication criteria? Is the manuscript technically sound, and do the data support the conclusions? The manuscript must describe methodologically and ethically rigorous research with conclusions that are appropriately drawn based on the data presented.

Reviewer #1: Yes

Reviewer #2: Yes

Reviewer #3: Yes

2. Has the statistical analysis been performed appropriately and rigorously?

Reviewer #1: Yes

Reviewer #2: Yes

Reviewer #3: Yes

3. Have the authors made all data underlying the findings in their manuscript fully available (please refer to the Data Availability Statement at the start of the manuscript PDF file)?

Reviewer #1: Yes

Reviewer #2: Yes

Reviewer #3: No

4. Is the manuscript presented in an intelligible fashion and written in standard English?

Reviewer #1: Yes

Reviewer #2: Yes

Reviewer #3: No

5. Review Comments to the Author

Reviewer #1: Reviewer comments for the manuscript “Evaluation of two red cell inclusion staining methods for assessing spleen function among sickle cell disease patients in North-East Nigeria”

The study investigates two different staining methods for assessing the current state of spleen function in apparently healthy controls and SCD patients. This is done by optimizing the staining methods, to keep the procedure as simple as possible. The stained RBCs are then counted for the presence and frequency of AI or HJB inclusions and the increased presence inclusion bodies is correlated to spleen function.

The study addresses a major problem in typically developing countries with considerable SCD prevalence, namely that the surveillance of the patient’s condition is hampered by the need for expensive equipment and knowhow. There these relatively simple methods could make it a routine task to follow disease development in patients over time, to pinpoint the need for treatment and increased observation, where possible.

The study is in general well performed, the writing is mostly clear and graphs and most pictures in a reasonably acceptable quality.

Major criticism and consideration: Both the two tested methods have previously been shown to be well-correlated and useful for easy testing spleen function, in similar studies, whereof at least two are cited in the current paper (Tham et al., 1996 and Corazza et al., 1996). In principle these two papers show the same as this study, however, the present study has the advantage that it has a bigger dataset for comparison and is performed in realistic settings for the assessment. Also, in favor is the lack of never papers on the subject and the simplified staining methods used, making it easier to employ in developing countries.

A study to follow up on the practical implementation of this method in developing countries would be very welcome.

The current study would benefit from being evaluated with regard to when to start treatment of the patients.

Maybe a short discussion of the findings with regard to a threshold value for the two staining methods, also seen in the light of previous studies, for when to initiate treatment, would be of general interest to the readers.

Comments to the text and figures:

1. Under Methods: “This was a prospective cross-sectional study conducted amongst children and adults with SCD in steady state between September 2020 and November 202; all patients…“. The second year is missing.

2. Under Method for AI count using silver stain: “All stained slides were mounted, as this improved the clarity of microscopy and reduced the effect of reflection observed with unmounted slides”. It would be nice to know what mounting medium was used or to have a recommendation. Also the coverslip may be of great importance and should be a #1.5 coverslip (thickness 170 µm) to give the best possible image quality.

3. Under Method for quantitative HJB count using May-Gruenwald Giemsa (MGG) stain: Were these slides not mounted? And if they were, with what and with what thickness coverslip etc.?

4. Under Method for AI count using silver stain:” The slides were stained using silver stain at 38C for 20 minutes” – The character used in 38⁰C is a 0 not a degree sign. Please revise.

5. Under Microscopy: “Microscopy was performed using a light microscope (Leica DM 750, equipped with a ICC50 E colour camera) at 400x and 1,000x magnification”. The magnification information should be left out as this does not make much sense (as the magnification varies with reproduction size). Therefore, scalebars in all images, and information on the objectives used is important. And again, be sure to infer Numerical Aperture (NA) as this tells something about resolution of the objective.

6. Just a comment for the next study - Under Microscopy: A nice test would have been to count the exact same pictures twice, to see the variation between the counts, as this would truly show the intraobserver variation. Two counts of different fields is bound to give different results even at 500 RBCs.

7. Under General characteristics: ” The median ages for the SCD group and controls were 11 years (range 1 - 45 years) and 12.0 years (range 1 - 32 years) respectively.”. Why is the digit left out of the SCD group (11), when it is stated the control age group is 12.0?

8. Under Identification of argyrophilic inclusion-positive (AI) and HJB red cells : “The AI inclusions were spherical, oval or irregular in shape, of variable sizes and occurred either as single or multiple inclusions within a red cell (Fig. 1)”. Could this also say: ”Anything appearing as dark spots or areas in the cells was considered to be AIs”? It just seems to be very random shapes and sizes.

9. Under Comparison of AI and HJB red cells count among the study population: It says several times that P=0.000. This cannot be, but most likely is just stated like this, since the value is lower than 0.001. Therefore it should either say the real value (eg. p=0.000023) or be shown as p<0.001 or p<0.0001.

10. In Discussion: “”…based on that first described by Tham et al in the United States (12).“. Maybe a little awkward to state that it happened in the United States…and remember to put . after al.

11. In Discussion: “In one study, the authors noted inclusions in 54.3% of red cells from 20 splenectomised individuals, using interference phase-contrast microscopy (21).”. This technique is not phase contrast, but Differential Interference Contrast (abbreviated DIC) and very different from phase contrast.

12. Under Discussion: “Kent al found increased red cells with inclusions…”, should be “Kent et al.”

13. Under Discussion: ”Whatever their nature, it appears the AI are found in increased numbers in SCD patients compared to HJB; the forma although more…”, should be “Whatever their nature, it appears the AI are found in increased numbers in SCD patients compared to HJB; the former, although more…”

14. Under Discussion: “Our data demonstrated a good correlation between the two sets of reading obtained with the AI test (R = 0.90; R2 = 0.82), in keeping with an earlier report (r=0.736; P<0.001)(12).” It is not clear to me, whether this is an intra- or interobserver variation you are considering here – I guess that it is intra? The earlier report mentioned is clearly stated as intraobserver values. Please clarify. And I do not see any examination of the interobserver variability in the Results section – has this been done? Maybe I just missed it.

15. Minor comment - Under Conclusions: “in larger studies may be required for the AI method before the generalizability of its efficacy in assessing spleen function in SCD can be ascertained.”. The z in generalizability indicates US spelling whereas other places, like under Microscopy “(Leica DM 750, equipped with a ICC50 E colour camera)”, colour is spelled as in the UK. Also in Discussion: “In one study, the authors noted inclusions in 54.3% of red cells from 20 splenectomised individuals” – again UK spelling. Please revise and unify.

Figures:

Fig. 1: The image should have a scalebar and information on the objective used (eg. 40x/0.65 or similar). Not clear if the arrows point to AIs or just cells. Consider white arrows that would stand out from the background better.

Fig. 2: Same comments as in Fig. 1 (scalebar and info needed), but here it is clearer that the arrows point to inclusions. Be sure to use same color and size of arrows in both figures.

Fig 3 and 4. Since many of the observations lie in the low percentage the graphs are very busy in those regions (especially for the HJB). Consider splitting up both in two showing observations in the 0-10 or 20% range and 10 or 20% to 100% on their own to better show the variation between observers.

S Fig 1: The image is in relatively bad quality, scalebar is missing, information about the microscope objective is needed (both magnification and NA) and it is unclear if the arrows point to AIs or just cells. Black arrows would be better visualized. Please revise accordingly.

S Figures 2-4: Consider putting them together in one figure, making it easier to compare the three.

S1 Appendix 1: Under Limitations: The text says “Kent et al found an increase in the percentage of red cells with autophagic”, but the Kent reference is not found in the list of papers and it should say a year, eg. Kent et al., 1998 (or similar). I later could find the paper in the general reference list. Please put it in the Limitations reference list too.

Reviewer #2: Comments to the authors

The manuscript by Ladu and colleagues reports two haematological methods to splenic function by comparing the proportions of red blood cells (RBC) containing Howell-Jolly body (HJB) and argyrophilic inclusions (AI)

RBC are known to contain various inclusions (HJB, argyrophilic inclusions, vacuoles, microorganisms during infections, etc.) over time. These inclusions are normally cleared by a healthy spleen through the pitting process. Their presence in circulating peripheral blood is therefore correlated to the spleen dysfunction (hyposplenia). Because gold standard methods such as scintigraphy or pitted RBC count require specific equipment not readily available in Africa, HJB and AI are suitable due to the only requirement of light microscope.

The manuscript is well written and the authors are to be commended of the clean presentation of figures, although images could be a little bit improved.

My comments and questions are the following

1. Another control group would strengthen the manuscript (e.g., splenectomized subjects, another hematological disorder).

2. In African context, do the authors take in account the malaria parasites? Do the authors have the malaria status of all the participants, as some can be asymptomatic and/or auto-medicated (pyknotic parasites)?

3. A clinical dimension is needed. A correlation between AI proportion and spleen status (size and not just presence/absence on ultrasonography) and/or patients’ other clinical data is important to validate the new method and assess the degree of spleen dysfuntion in this population.

4. A little bit more details on AI compositions (nucleic acid? Proteins?) are needed in the introduction even though discussed later (discussion).

5. In Tham et al., hematopathology, 1996: they reported a greater sensitivity of argyrophilic RBC count over HJB as a test for splenic function, why not look for another marker of spleen function (pocked even though not accessible, it is not impossible for such an impacting study). It would be great to see the correlation with pitted RBC specifically, at least for some participant.

6. What the authors can propose to improve the working solution stability, as it deteriorates after about 3 minutes of preparation?

7. Do the authors think that 500 RBC per smear is enough to reasonably assess the proportion of AI and 400 RBC per smear for HJB? Depending on the field, this count can drastically change.

8. The authors elaborate about intra-observer variation, what about inter-observer variation?

9. If argyrophilic RBC proportions are correlated to pocked RBC proportions (as reported in the discussion), what that tells about the contain of vacuoles in pocked RBC specifically knowing that some vacuoles are empty?

Minor points

- Number lines to facilitate the reviewer comments and suggestions

- The first sentence of the methods “study participant”: I guess the enrolment is between “September 2020 and November 2022?”

- In methods “Blood smears”: what explains the discrepancy between the fixing solution in the manuscript and that of Tham et al.,1996 “Fix a thoroughly air-dried blood smear or a Wright-stained smear for 3 minutes in a solution made by diluting 150 mL of formalin (37% formaldehyde) to 500 mL with 95% ethanol”?

- On figures 5, 6a and 6b, horizontal lines (dashed or not) delineating variations must be replaced accordingly.

Reviewer #3: I have attached my detailed comments for the authors.

I would like to use this space to say that the manuscript requires English Language revising.

My comments about the data presentation and scientific content of the work is in the attached document.

I will be very happy to reread the articles after the reviews.

6. PLOS authors have the option to publish the peer review history of their article (what does this mean?). If published, this will include your full peer review and any attached files.

**Do you want your identity to be public for this peer review?** For information about this choice, including consent withdrawal, please see our Privacy Policy.

Reviewer #1: No

Reviewer #2: **Yes: **Abdoulaye Sissoko

Reviewer #3: **Yes: **Sara El Hoss

---

## [Decision Letter · Decision Letter 1]

24 Apr 2023

Evaluation of two red cell inclusion staining methods for assessing spleen function among sickle cell disease patients in North-East Nigeria

PGPH-D-22-02120R1

Dear Dr Ladu,

We are pleased to inform you that your manuscript 'Evaluation of two red cell inclusion staining methods for assessing spleen function among sickle cell disease patients in North-East Nigeria' has been provisionally accepted for publication in PLOS Global Public Health.

Best regards,

Debjani Paul

Academic Editor

The authors have incorporated the suggestions of the reviewers as best as possible. It has resulted in significant improvement of the revised manuscript.

Reviewer Comments (if any, and for reference):

Reviewer's Responses to Questions

**Comments to the Author**

1. If the authors have adequately addressed your comments raised in a previous round of review and you feel that this manuscript is now acceptable for publication, you may indicate that here to bypass the “Comments to the Author” section, enter your conflict of interest statement in the “Confidential to Editor” section, and submit your "Accept" recommendation.

Reviewer #2: All comments have been addressed

Reviewer #3: All comments have been addressed

2. Does this manuscript meet PLOS Global Public Health’s publication criteria? Is the manuscript technically sound, and do the data support the conclusions? The manuscript must describe methodologically and ethically rigorous research with conclusions that are appropriately drawn based on the data presented.

Reviewer #2: Yes

Reviewer #3: Yes

3. Has the statistical analysis been performed appropriately and rigorously?

Reviewer #2: Yes

Reviewer #3: Yes

4. Have the authors made all data underlying the findings in their manuscript fully available (please refer to the Data Availability Statement at the start of the manuscript PDF file)?

Reviewer #2: Yes

Reviewer #3: Yes

5. Is the manuscript presented in an intelligible fashion and written in standard English?

Reviewer #2: (No Response)

Reviewer #3: Yes

6. Review Comments to the Author

Reviewer #2: (No Response)

Reviewer #3: I think the authors for the responses they provided to my comments and to the comments of the other reviewers. The manuscript is now very well written and the data are very interesting.

7. PLOS authors have the option to publish the peer review history of their article (what does this mean?). If published, this will include your full peer review and any attached files.

**Do you want your identity to be public for this peer review?** For information about this choice, including consent withdrawal, please see our Privacy Policy.

Reviewer #2: **Yes: **Abdoulaye Sissoko

Reviewer #3: **Yes: **Sara El Hoss
